# Isolation, Identification, and Whole Genome Analysis of Chicken Infectious Anemia Virus in an Outbreak of Disease in Adult Layer Hens

**DOI:** 10.3390/vetsci10070481

**Published:** 2023-07-23

**Authors:** Yueyan Zeng, Hui Zhang, Huanrong Zhang

**Affiliations:** 1College of Animal and Veterinary Sciences, Southwest Minzu University, Chengdu 610041, China; zengyueyanstone@gmail.com (Y.Z.); dkyzhanghui@163.com (H.Z.); 2Key Laboratory of Veterinary Medicine, Universities in Sichuan, Chengdu 610093, China

**Keywords:** chicken infectious anemia virus, isolation and identification, pathogenicity, whole genome, bioinformatics analysis

## Abstract

**Simple Summary:**

Chicken infectious anemia (CIA) is a prevalent immunosuppressive disease that affects poultry and is caused by the chicken infectious anemia virus (CIAV). It primarily induces aplastic anemia in chickens and usually leads to subclinical infections in adult chickens. Recently, a severe disease outbreak occurred on a large-scale layer hen farm in Guangxi Province, China, housing approximately 1,000,000 20-week-old hens. The outbreak resulted in an alarming average of 550 daily deaths over a span of 10 days. Clinical symptoms prompted laboratory investigations for common avian diseases and bacterial infections in the affected layer hens. The test results revealed the presence of CIAV exclusively, strongly suggesting its significant role as the causative pathogen in this outbreak. Consequently, our study aimed to investigate the etiology and molecular characteristics of the CIAV strains present on this farm. Through our research, we identified three prevailing strains of CIAV, and subsequent animal infection experiments confirmed the high pathogenicity of the GX21121 strain. These strains caused characteristic lesions and resulted in 100% mortality in 1-day-old specific-pathogen-free (SPF) chicks from 4 to 29 days post-infection (dpi). Additionally, we conducted a genetic analysis to determine the evolutionary relationships among the strains and explore the possibility of recombination. In summary, our study has provided valuable insights into the pathogenicity and genomic characteristics of the recently isolated CIAV strains. This information can serve as a reference for the prevention, control, and traceability of CIA in poultry farms.

**Abstract:**

Chicken infectious anemia (CIA) poses a significant threat to the chicken industry in China. Due to its non-specific symptoms, the disease is often overlooked. This study aimed to conduct a comprehensive analysis of the etiology and pathology of CIA in Guangxi Province, China. Three strains of the chicken infectious anemia virus (CIAV) were isolated from liver samples of diseased 20-week-old chickens. The complete genomes of these strains were sequenced, and experiments on specific pathogen-free (SPF) chicks revealed that the GX21121 strain exhibited high virulence. Histopathological examination of the deceased chickens showed liver cell necrosis, fibrous serous exudation, inflammatory cell infiltration, hemorrhage in liver tissues, and congestion in lung and renal tissues. Phylogenetic analysis of the genome revealed that the three strains had a close genetic relationship to the Heilongjiang wild-type strain (GenBank KY486144). The genetic evolution of their VP1 genes indicated that all three CIAV isolates belonged to genotype IIIc. In summary, this study demonstrated the genomic diversity of three CIAV strains in adult layer hens. The isolation and characterization of the GX21121 strain as a highly virulent isolate provide valuable information for further investigations into the etiology, molecular epidemiology, and viral evolution of CIAV.

## 1. Introduction

Chicken infectious anemia (CIA) is a viral disease caused by the chicken infectious anemia virus (CIAV) and primarily affects young chicks. It is characterized by aplastic anemia, pale bone marrow, lymphoid tissue atrophy, and compromised immune function [1,2]. CIAV is a non-enveloped, icosahedral symmetrical virus particle belonging to the Anelloviridae family, with a diameter ranging from 25 nm to 26.5 nm and a genome size of approximately 2.3 kb [3,4,5]. The first isolation of CIAV was reported in Japan in 1970 from a contaminated vaccine [6]. Although chickens are the natural host of CIAV, the virus has also been detected in the feces of humans, mice, dogs, and other birds [7,8,9]. To date, CIA has been reported in major poultry-raising countries worldwide. The disease primarily affects chicks between 1 and 3 weeks of age [10,11], while adult chickens generally experience subclinical infections. Transmission of CIAV occurs through vertical and horizontal routes, such as oral–fecal contamination, and occasionally through contaminated vaccines [12,13]. CIAV can propagate in chicken embryos and lymphoblastoid cell lines but not in chicken embryo fibroblasts, chicken kidney cells, or other primary cells [14]. The MDCC-MSB1 cell line has been widely employed for CIAV isolation in previous studies [15]. However, different CIAV isolates exhibit varying sensitivities and replication rates on MDCC-MSB1 cell lines [16].

The genome of CIAV encodes three viral proteins: the capsid protein (VP1, 51.6 kDa), the viral backbone protein (VP2, 24 kDa), and the apoptosis-inducing protein (VP3, 13.6 kDa) [17,18]. VP1 functions as a structural protein that, in conjunction with VP2, generates a neutralizing epitope. However, research has also shown that VP1 contains a neutralization epitope, although variations exist among mutant strains, and the underlying mechanisms are yet to be fully explained [19,20,21]. Acting as the backbone protein, VP2 performs multiple roles in the virus replication process [22,23], while VP3 induces apoptosis in chicken thymic lymphoblasts and primitive hematopoietic cells, leading to severe clinical symptoms such as immunosuppression, hemorrhage, and anemia [24]. The VP1 gene is involved in viral replication and pathogenicity, and extensive studies have been conducted on its sequence variability [25,26,27]. Comparatively, the VP2 and VP3 genes exhibit higher conservation than VP1 [28]. Therefore, the VP1 gene serves as the primary source of variation among CIAV strains. Based on nucleotide sequence variations in the VP1 gene, four distinct genotypes (I, II, III, IV) have been identified worldwide [29,30]. Furthermore, studies have revealed the occurrence of recombination events in the CIAV genome, potentially leading to the emergence of new genotypes [31].

In Guangxi Province, China, a severe outbreak occurred on a large-scale 20-week-old layer hen farm, resulting in an average of 550 deaths per day over a period of 10 consecutive days. The outbreak was confined to Shed 7, while the remaining nine sheds remained unaffected. The sheds were evenly spaced 60 m apart, and uniform management practices were implemented across all sheds. Prior to the onset of production, the chickens were grouped and reared separately for a duration of 60 days. All chickens underwent the same routine immunization program for commercial layer farms before grouping. Following grouping, each group received the same immunization program, with no specific immunization against CIAV. The objectives of this study were to isolate and identify CIAV strains, investigate their pathogenicity and genomic characteristics, enhance the understanding of the etiology and molecular biology of CIAV isolates in China, and provide valuable information for the prevention, control, and traceability of CIA.

## 2. Materials and Methods

### 2.1. Sample Collection

In December 2021, a total of 50 liver samples were collected from 20-week-old sick layer hens on a farm in Guangxi Province, China.

### 2.2. Reagents and Materials

*Taq* DNA polymerase and dNTPs were purchased from TaKaRa Company; DL 2000 DNA Marker was purchased from Shanghai Sangon Bioengineering Technology Service Co., Ltd.; agarose was purchased from OXOID Company; 1 × TAE electrophoresis solution, EB nucleic acid dye, PBS solution (pH 7.4, 0.01 M, containing 1000 units/mL penicillin), and virus genome DNA/RNA extraction kit were purchased from Tiangen Biochemical Technology Co., Ltd., Beijing, China; Gel Extraction Kit D2500 was purchased from OMEGA; pMD 19-T vector was purchased from Baobio Engineering Co., Ltd. (Dalian, China); DH5α competent cells were purchased from Thermo Scientific; SPF chicken embryos were purchased from Sichuan Huapai Bioengineering Group Co., Ltd., Jianyang, China; MDCC-MSB1 cells were preserved by our laboratory.

### 2.3. Primer Design and Synthesis

According to the published CIAV genome sequence (GenBank KY486149.1), Oligo7 software was used to design and synthesize four pairs of primers (Table 1) for detecting CIAV and amplifying the whole genome. Primers were synthesized by Sangon Bioengineering Co., Ltd. (Shanghai, China).

### 2.4. Sample Handling

Samples were collected and processed for routine Polymerase Chain Reaction (PCR) testing, following the pooling of every 5 samples. The PCR analysis aimed to identify common poultry viruses, including Marek’s disease virus (MDV), avian influenza virus (AIV), infectious bursal disease virus (IBDV), infectious bronchitis virus (IBV), reticuloendotheliosis virus (REV), chicken infectious anemia virus (CIAV) [32,33,34,35]. Bacterial isolation was also conducted using a Luria–Bertani agar plate, MacConkey agar plate, and Blood agar plate. among the samples examined, only CIAV was detected as positive. To further investigate CIAV, fifteen sick chickens exhibiting typical clinical symptoms associated with CIA were selected. Based on the most prominent PCR amplification bands and the absence of detection for other common poultry diseases, a subset of six samples was chosen for individual CIAV PCR testing. These selected samples were stored at −80 °C for subsequent experiments.

The six diseased chicken livers were individually labeled as samples 1 to 6. After being weighed and cut into small pieces, they were thoroughly mixed with PBS solution (pH 7.4, 0.01 M, containing 1000 units/mL penicillin). The mixture was homogenized to obtain a tissue suspension with a concentration of 200 g/L. The tissue suspension was then frozen at −80 °C for 30 min and subsequently placed in a constant temperature water bath at 37 °C. The suspension was vigorously shaken until complete dissolution, and this freeze–thaw process was repeated three times to ensure proper disruption of the cells. After the freeze–thaw cycles, the suspension was centrifuged at 5330× *g* rpm and 4 °C for 5 min. Following centrifugation, 200 μL of the supernatant was collected, and DNA extraction from the viral genome was performed using a DNA/RNA extraction kit. The extracted DNA from each sample was labeled accordingly as 1 to 6 and stored at −80 °C for future use.

#### 2.4.1. PCR Detection

The CIAV primers ①–④ in Table 1 were used to perform PCR amplification of the DNA extracted as described in Section 2.4. In total, the 20 μL PCR reaction system included extracted DNA 1 μL, Taq enzyme 10 μL, ddH2O 7 μL, upstream and downstream primers 1 μL each; reaction condition was as follows: pre-denaturation at 95 °C for 5 min; denaturation at 94 °C for 30 s, annealing at 50.5 °C for 30 s, extending at 72 °C for 30 s, 30 cycles; extending at 72 °C for 7 min; storing at 16 °C to end the reaction. 5 μL of the PCR product was electrophoresed in a 1.5% agarose gel at 120 V and 200 mA for 28 min.

#### 2.4.2. Virus Isolation and Identification

The samples initially tested positive for CIAV using PCR as described in Section 2.4.1. To eliminate potential viral contaminants, an equal amount of chloroform was added to the supernatants of the samples under aseptic conditions. The mixture was incubated for 15 min. Subsequently, to remove potential bacterial contamination, the supernatants were exposed to a 70 °C water bath for 5 min. The treated virus suspensions were then inoculated into the yolk sacs of three 6-day-old SPF chicken embryos, with each chicken embryo receiving a volume of 0.15 mL. At 13 dpi, the SPF chicken embryos were dissected aseptically to collect the allantoic fluids, liver, spleen, thymus, and bursa of Fabricius.

The collected tissues were thoroughly homogenized to create a tissue suspension with a concentration of 200 g/L. After homogenization, the mixture was centrifuged at 2665× *g* rpm and 4 °C for 20 min. The resulting supernatant was carefully collected, and to ensure purity, it was filtered through a 0.45 μm filter. This process yielded the isolated virus stock, which was ready for further analysis and experimentation.

The isolated virus stock was subjected to three successive passages using the aforementioned method. After each passage, the harvested virus stock was analyzed using PCR. If all the PCR results for the harvested virus stock were positive, it indicated the successful isolation of a CIAV strain.

#### 2.4.3. MDCC-MSB1 Cells Inoculation of CIAV Isolate

The isolated viral suspension was inoculated onto 5 × 10^3^~5 × 10^5^ MDCC-MSB1 cells and cultured for 4 days at 37 °C under 5% CO_2_ condition to obtain the first passage of cell culture virus. DNA was extracted from the harvested cells and culture media for PCR detection. The CIAV-positive culture was continuously passaged up to the 5th passage for observing cytopathic effects (CPE). If CPE was observed or PCR results were positive, the virus strain was deemed to be adapted to the MDCC-MSB1 cell line.

### 2.5. Whole Genome Amplification of CIAV Isolates

Using the four pairs of primers listed in Table 1, we performed PCR amplification on the chicken embryo isolates to obtain four partially overlapping sequences with estimated lengths of 1010 bp, 751 bp, 1055 bp, and 540 bp. The amplification reactions were carried out in a 25 μL reaction system as described in Table 2. The reaction mixture consisted of 2 μL of total supernatant DNA, 12.5 μL of Taq enzyme, 8.5 μL of ddH_2_O, and 1 μL each of the upstream and downstream primers. The amplified products were visualized by electrophoresis on a 1.5% agarose gel, and the PCR amplification products were then extracted using the OMEGA Gel Extraction Kit D2500. Subsequently, the purified products were ligated into the pMD19-T vector for sequencing. The obtained sequencing results were analyzed using DNA STAR software SeqMan Pro v11.1.0 (59), 419 which allowed for the assembly of the sequences to obtain the full-length genome of the CIAV strain isolated from the chicken farm.

### 2.6. Experimental Infection of 1-Day-Old SPF Chicks with CIAV

Eighteen 1-day-old SPF chicks were intramuscularly inoculated with 0.3 mL (10^4^ EID50) of the chicken embryo passage virus solution per chick in the leg, while an additional six chicks were inoculated with equal doses of saline as a control. The two groups of chicks were housed separately in appropriate conditions to ensure their comfort. Throughout the duration of the experiment, any deceased test chicks were promptly dissected and subjected to testing. Organs exhibiting lesions were carefully selected, fixed in 10% formalin, and subsequently embedded in paraffin for the preparation of pathological histological sections. These sections were then stained using hematoxylin and eosin (HE), and images were captured using a 3DHISTECH (Hungary) Pannoramic 250 digital section scanner to observe any histopathological changes.

### 2.7. Genome-Wide Bioinformatics Analysis

#### Genome-Wide Genetic Evolutionary Analysis

In this study, a total of 75 reference strains of CIAV isolated from different periods and regions were selected based on their full sequences, available in GenBank (see Appendix A). The whole genome sequences obtained through amplification in our study were aligned with these reference strains using MEGA-X software, employing the Clustal W algorithm. The genetic evolutionary tree was constructed using the Neighbor-Joining (NJ) method, with 1000 bootstrap replicates. To determine the genotype of the CIAV strain from the chicken farm under investigation, the typing method proposed by Ducatez, M.F. et al. [29] was employed. To explore the possibility of recombination events among the selected sequences, we utilized RDP 4.0 software [36]. Multiple methods, including RDP, Geneconv, Bootscan, Maxchi, Chimaera, Siscan, and 3Seq, were employed with a *p* value adjusted to 0.05. The recombination events identified were further validated using SimPlot v.3.5.1 [37].

## 3. Results

### 3.1. Pathological Changes

The dead layer hens from the farm showed depression and pale comb (Figure 1a). Postmortem examination showed scattered subcutaneous hemorrhage and pale muscles (Figure 1b), enlarged liver, and liver surface petechiae and necrotic foci (Figure 1c).

### 3.2. PCR Test Results of the Samples

All the selected six liver samples, which exhibited pathological changes, tested positive for CIAV. The electrophoresis analysis revealed a band size of approximately 500 bp, which corresponded to the expected target band (Figure 2a).

### 3.3. Virus Isolation Results

After three consecutive passages in SPF chicken embryos, PCR testing of the chicken embryo yolk sac culture virus nucleic acid confirmed three positive results for CIAV (Figure 2b), indicating the successful isolation of three CIAV isolates. These isolates were named GX21121, GX21122, and GX21123. Among these isolates, strain GX21121 exhibited persistent liver lesions and specific PCR bands even after three additional passages. Therefore, strain GX21121 was selected for further investigation of its pathogenicity.

### 3.4. MDCC-MSB1 Culture Results

After incubating the MDCC-MSB1 cells with CIAV strain GX21121 for 5 days, the PCR test consistently yielded negative results. This trend continued even after four consecutive passages. The experiment was repeated five times, and each time the PCR test yielded negative results. Based on these findings, it can be concluded that the GX21121 strain isolated from the SPF chicken embryo yolk sac is not adapted to the MDCC-MSB1 cell line.

### 3.5. Whole Genome Amplification Results

The results of PCR amplification of the four fragments are shown below (Figure 2c). They were consistent with expectations. In this study, three CIAV complete gene sequences of 2298 kb referring to CIAV isolates GX21121, GX21122, and GX21123 (GenBank accession numbers: OQ267594, OQ267595, OQ267596) were obtained after sequencing and splicing. A genomic schematic diagram of the GX21121 strain was constructed based on gene annotation information to illustrate the basic genomic features of the isolates (Figure 2d).

### 3.6. CIAV GX21121 Pathogenicity to One-Day-Old SPF Chicks

After inoculating the experimental SPF chicks with CIAV GX21121, a total of 18 chicks were monitored. Two chicks died at 7 dpi, five chicks died at 14 dpi, and the remaining 11 chicks died successively between 15 and 29 dpi. No deaths were observed in the negative control (NC) group. When compared to the NC, the deceased chicks infected with CIAV exhibited significant clinical signs of depression and rough feathers at 7 and 14 dpi. Varying degrees of scattered subcutaneous hemorrhagic spots were observed. Additionally, the infected chicks showed pale muscles, yellow staining of the liver with white necrotic foci at the edges, swollen and hemorrhaged kidneys at 7 dpi, pale kidneys at 14 dpi, swollen spleen with white necrotic foci on the surface at 7 dpi, and necrosis at the edge of the spleen at 14 dpi. The 7 dpi chicks displayed thymus enlargement and hemorrhage, whereas the 14-day-old dead chicks showed thymus atrophy. Selected significant changes between the experimental group (EG) and the negative control (NC) are presented below (Figure 3a–h).

Microscopic examination of histopathological sections from the 7 dpi deceased chicks revealed the absence of clear lobular division and disordered arrangement of hepatic cords of liver. Extensive hepatocellular necrosis was observed, characterized by cytoplasmic shrinkage or dissolution, nuclear pyknosis, and disappearance. Patchy hepatocyte necrosis resulted in the formation of necrotic foci. Inflammatory cells, primarily heterophils, infiltrated the area, accompanied by fibrino-serous exudation. The necrotic foci were also associated with bleeding, visible red blood cell aggregation, and the accumulation of light-staining fluid around their edges. No significant inflammatory cell infiltration was observed in the liver sinusoids. The intrahepatic portal area, interlobular arteries, interlobular veins, and interlobular bile duct structures remained intact, with no apparent fibrous tissue proliferation or inflammatory cell infiltration. In the lung tissue, the surface was covered with pleura, without visible lobular division. The connective tissue of the pleura extended deep into the lung parenchyma, dividing it into several pulmonary lobes. The bronchial structure at all levels of lung tissue appeared normal, with relatively neat arrangement of the bronchial ciliated epithelium. Fibrin-like exudate was present inside the bronchi, and the bronchial lumen contained a light-staining substance. The structure of the third-level bronchi, pulmonary lobules, and pulmonary capillaries was clear. Inflammatory cells, mainly segmented or rod-shaped heterophils, infiltrated into the loose connective tissue around the blood vessels. Some fibrous tissue proliferation was observed in the lung tissue, with elongated nucleus-shaped fibroblasts. A significant aggregation of red blood cells was observed, while no other notable pathological changes were observed. In the kidney tissue, the renal capsule remained intact, and the boundary between the cortex and medulla was relatively clear. The structure of the cortical glomeruli was complete and clear, with no obvious degeneration or necrosis. The renal tubules were arranged relatively regularly. Some renal tubular degeneration and necrosis were observed, with degenerated and necrotic renal tubular epithelial cells exhibiting cytoplasmic dissolution, nuclear shrinkage, and disintegration, resulting in the disappearance of renal tubule structures in severely affected areas, forming necrotic foci accompanied by fibrous tissue proliferation. Inflammatory cells, including heterophils and lymphocytes, infiltrated the area, along with elongated nucleus-shaped fibroblasts. No significant inflammatory infiltration or fibrosis was observed in the interstitium (Figure 3i–k). CIAV nucleic acids were detected positive in all experimental infected chicks.

### 3.7. Genome-Wide Bioinformatics Analysis

The phylogenetic analysis of the complete genomes of the isolated strains and reference strains revealed close relationships among the three isolated strains. GX21122 and GX21123 formed a distinct minor branch, indicating a closer evolutionary relationship between them compared to GX21121 (Figure 4A). The reference strains that exhibited the closest genetic relationship were obtained from the northeastern region of China (GenBank KY486144, KY486155, KY486149). Nucleotide homology analysis showed a high level of similarity among the isolated strains, with the highest homology of 99.74% observed with the reference strains (GenBank KY486144), which is consistent with the phylogenetic tree based on complete genomes. The VP1 phylogenetic tree demonstrated a tight clustering of the VP1 sequences of the three isolated strains, along with the Heilongjiang strain (GenBank KY486144) and the Jilin strain (GenBank KY486155), all belonging to genotype IIIc (Figure 4B). Analysis of the VP1 amino acid sequences revealed the presence of glutamine at position 394 in all three isolated strains, which is considered a characteristic feature of virulent CIAV strains [38].

Recombination analysis using RDP 4.0 software indicated that GX21122 and GX21123 were potential recombinant strains involved in the same recombination event. The major parent was the Guangxi strain (GenBank MN103406) of genotype IV, while the minor parent was not detected. The predicted recombination region was identified between nucleotide positions 1790–2210. This recombination event was supported by seven out of nine methods used in RDP 4.0 (Table 3 and Figure 4C,E). The SimPlot analysis confirmed these results, showing a potential fragment recombination event of GX21122 and GX21123 in the region of 1679–2132 nt (Figure 4D,F). No potential recombination signal was detected for GX21121.

## 4. Discussion

This study investigates an outbreak of disease that occurred in a 20-week-old laying hen farm in Guangxi Province, China. The affected hens exhibited clinical symptoms and pathological changes consistent with anemia, including depression and pale combs and muscles. We successfully isolated and identified three strains of CIAV, namely GX21121, GX21122, and GX21123, using SPF chicken embryo yolk sac inoculation. The full genome sequences of these strains were obtained through PCR amplification and splicing, and their corresponding GenBank accession numbers are OQ267594, OQ267595, and OQ267596. Through pathogen isolation, identification, and comprehensive genome analysis, we confirmed the presence of CIAV on this particular farm.

Epidemiological surveys have shown that the prevalence of CIAV infection in chicken flocks in China ranges from 40% to 70%. It is particularly common in chickens aged 1 to 3 weeks, as young chicks are highly susceptible [10,11]. However, it is noteworthy that this outbreak occurred in 20-week-old laying hens, which are generally considered less susceptible to CIAV infection. CIAV infection can induce immunosuppression in chickens, making them more susceptible to other pathogens. Co-infections involving CIAV and other pathogens have been documented in clinical cases [39,40,41], suggesting a possible association with this outbreak. However, laboratory testing conducted on the affected chickens in this study only detected the presence of CIAV. Considering the clinical symptoms observed, it is a possibility that CIAV was the primary causative pathogen of this outbreak. It is important to note that the limited scope of pathogen testing should be taken into consideration, but these findings emphasize the potential impact of a CIAV outbreak on poultry farms. The results suggest that CIAV-induced immunosuppression can manifest in various clinical signs and lesions within the flock, which may vary depending on the age of the chickens.

No CPE were observed in MDCC-MSB1 cells infected with the GX21121 strain, even after repeating the experiment five times. Additionally, consistent negative results were obtained from PCR testing, indicating that the isolate did not exhibit adaptability to MDCC-MSB1 cells. This observation is consistent with previous reports that have highlighted variations in CIAV infectivity among different subtypes of MSB1 cells [15]. Different CIAV strains display differential sensitivity depending on the MSB1 cell subtype. For example, the CIA-1 strain [42] may not replicate in MSB1-L cells but shows limited replication in MSB1-S cells, while both cell subtypes are susceptible to the CUX-1 strain [17]. The underlying reasons for these differences are not yet fully understood. Our analysis primarily focused on the preliminary assessment of GX21121′s adaptability to MDCC-MSB1 cells. However, subsequent animal pathogenicity experiments confirmed the significant virulence of GX21121, indicating that the invasiveness of CIAV in cells is not strongly correlated with its pathogenicity. It is important to note that definitive conclusions regarding cell cultivation conditions could not be drawn from this study. Nevertheless, these findings provide preliminary insights into investigating the differential infectivity of CIAV in MSB1 cells.

All 1-day-old SPF chicks infected with the GX21121 strain experienced mortality within 29 days, with acute deaths observed at 7 dpi. These infected chicks displayed clinical symptoms such as depression and showed macroscopic and microscopic pathological changes, including scattered subcutaneous hemorrhagic lesions, liver necrosis foci, and severe anemia. It is important to note that GX21121 was isolated from a flock that had not been vaccinated against CIAV. Considering that only one shed of the poultry farm experienced morbidity and mortality, with no previous outbreaks of other infectious diseases, the likelihood of the outbreak being caused by feed or husbandry management factors is considered low. The large scale of the poultry farm and separation of groups of birds subjected to the same immunization procedures, but potentially differing vaccine batches, suggest the possibility of the virus being introduced through vaccines, and that, because of the large scale of the operation, variations in vaccine production batches occurred, which may have been the possible source of the virus, although efforts were made to ensure that each group received the same batch [2,4]. Further analysis of the complete genome sequence revealed a close genetic relationship between the isolated strains and three strains from the northeastern region of China, all belonging to genotype IIIc. Recombination analysis identified reliable recombination signals between GX21122 and GX21123 within nucleotide positions 1679–2132. The parental strains involved in the recombination event were genotype IV field strains prevalent in Guangxi. This recombination event occurred within amino acid positions 282aa–433aa of the VP1 protein. Interestingly, previous reports have highlighted significant differences between genotypes IIIc and IV in the VP1 amino acid sequence, particularly within positions 294aa–448aa [43]. Therefore, it is highly likely that recombination occurred between the parental strains in this region, resulting in the emergence of a new strain with an altered genetic profile. Multiple recombination events were detected during the analysis, highlighting the importance of recombination as a crucial mechanism for generating new CIAV strains and genotypes. Despite the overall conservation of CIAV sequences, it is important to acknowledge evolutionary trends, as this disease outbreak may indicate the onset of evolutionary changes in the infectivity characteristics of the virus.

Indeed, vaccination is an effective measure to control CIAV and prevent outbreaks of the disease. The outbreak described in this study can be attributed to the lack of vaccination against CIAV on the affected poultry farm. The study aimed to analyze the genetic diversity and evolution of the three CIAV strains isolated from a large-scale layer hen farm in Guangxi Province, China. The genomic characteristics of the isolated strains were examined, and the pathogenicity of the GX21121 isolate was clarified. By analyzing the genetic diversity and evolution of the isolated strains, the study provides important insights and references for the prevention, control, and traceability of CIA. Understanding the genomic characteristics of the strains and clarifying the pathogenicity of GX21121 can aid in developing effective control strategies, including vaccination programs, to prevent future outbreaks and mitigate the impact of the disease on poultry farms. The study’s findings serve as valuable materials and references for stakeholders involved in the management and control of CIA in poultry populations.

## 5. Conclusions

In this study, an outbreak of disease in a 20-week-old laying hen farm in Guangxi Province, China was investigated. The study suggested that the clinical symptoms and pathological changes could be associated with CIAV infection of the chickens. Three CIAV strains were isolated from the affected farm and subjected to comprehensive analysis. The whole genome analysis of the isolated strains provided insights into their genetic characteristics and evolutionary relationships. The study also examined the possibility of recombination events among the strains. Additionally, an animal pathogenicity experiment confirmed the high virulence of the GX21121 isolate, as evidenced by the development of typical CIA lesions and the 100% mortality rate observed in 1-day-old SPF chicks. The findings of this study contribute valuable information to the understanding of CIAV pathogenicity, genomic characteristics, and evolution. This information can be utilized for further research on the etiology, molecular epidemiology, and viral evolution of CIAV. The study serves as a foundation for future studies aimed at improving prevention, control, and management strategies for CIAV outbreaks in poultry farms.

## Figures and Tables

**Figure 1 vetsci-10-00481-f001:**
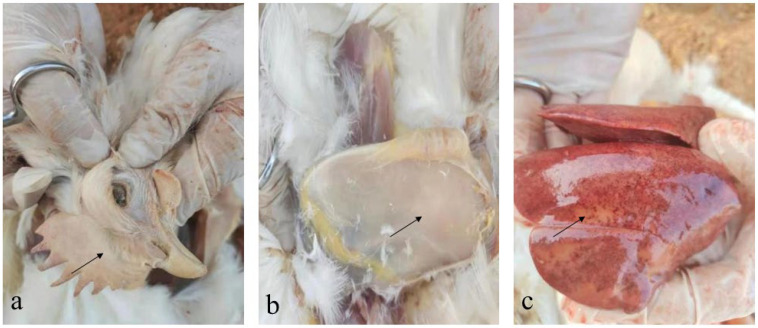
Typical lesions of sick and dead chickens. (**a**) pale comb. (**b**) pale breast muscles. (**c**) liver diffused hemorrhage and hepatic necrosis.

**Figure 2 vetsci-10-00481-f002:**
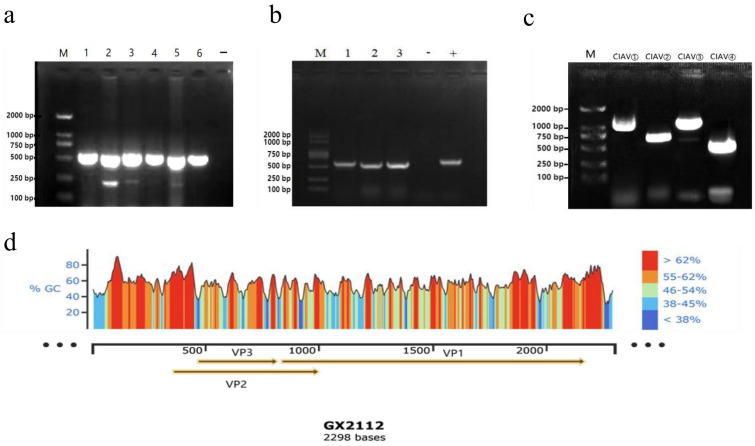
(**a**) PCR detection results of the 6 liver samples suspected to be CIAV positive. Note: M: DNA Marker (DL 2000); 1~6: clinical samples; −: negative control. (**b**) PCR detection results of three CIAV isolates. Note: M: DNA Marker (DL 2000); 1~3: GX21121, GX21122, and GX21123 viruses isolated from 3 liver samples; −: negative control; +: positive control. (**c**) Four fragments of GX21121 gene amplification electrophoresis results. (**d**) CIAV GX21121 DNA simulation map by SnapGene software (version 4.2). Note: M: DNA Marker (DL 2000); CIAV①~④: four amplified fragments with 4 primers; the yellow arrow part is the coding region, encoding 3 proteins: VP1, VP2, VP3.

**Figure 3 vetsci-10-00481-f003:**
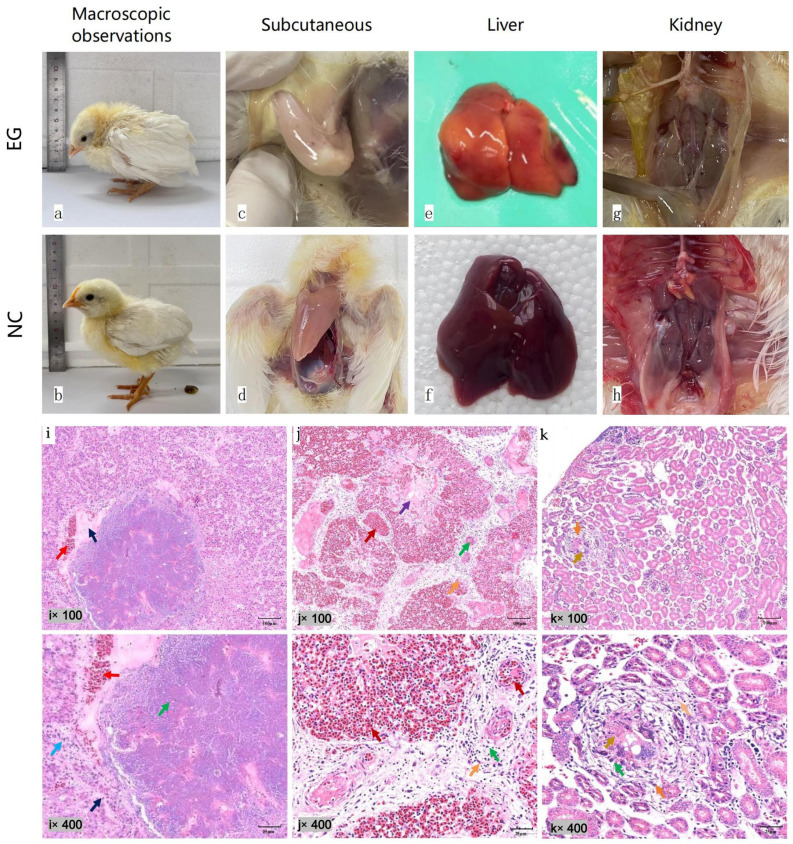
(**a**–**h**) Pathological examination of experimental SPF chicks revealed significant differences between the infected chicks and the NC. The infected chicks exhibited notable signs of depression and rough feathers compared to the NC (**a**,**b**). The EG displayed scattered subcutaneous hemorrhagic spots and pale muscles, which were not observed in the NC (**c**,**d**). The infected chicks also showed a yellowish appearance in the liver with surface foci of necrosis, while the NC did not exhibit such lesions (**e**,**f**). The kidneys of the infected chicks appeared pale, without any apparent lesions observed in the NC (**g**,**h**). Note: dpi refers to days post-infection, EG refers to the experimental group, and NC refers to the negative control. (**i**–**k**) Microscopic examination of the histopathological changes in the liver tissue of the 7 dpi chicks (**i**) liver cell necrosis, fibrinous serous exudation, inflammatory cell infiltration, and hemorrhage in some liver tissues. (**j**) Lung tissues exhibited fibrinoid exudation and inflammation. (**k**) Renal sections showed renal cell infiltration, fibrous tissue hyperplasia and congestion, renal tubular degeneration and necrosis, inflammatory cell infiltration, and fibrous tissue hyperplasia.

**Figure 4 vetsci-10-00481-f004:**
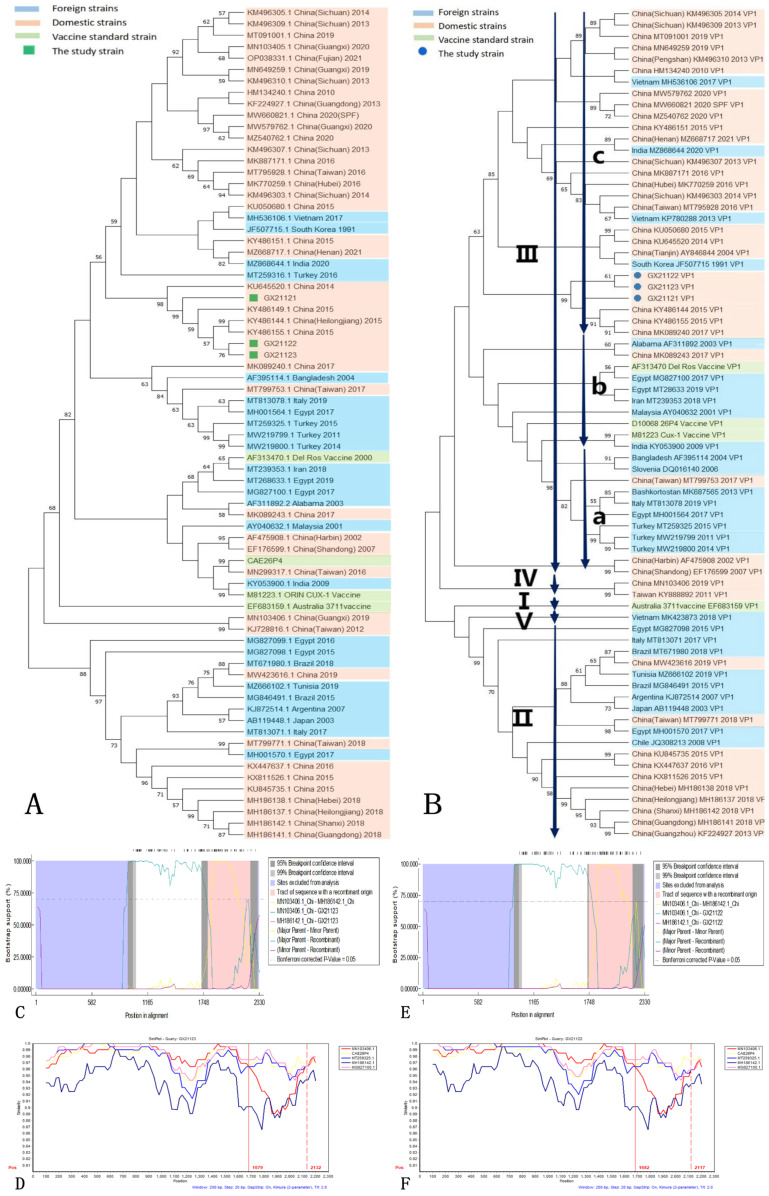
(**A**) Genome-wide phylogenetic tree. (**B**) Phylogenetic tree of VP1. (**C**,**E**) The result of the bootstrap algorithm in RDP recombination analysis for GX21122 and GX21123; the pink area represents the tract of sequence with a recombinant origin indicating the predicted region where recombination may have occurred. The intertwined area of multiple sequences represents the predicted potential recombination site or fragment. (**D**,**F**) The result of the Simplot algorithm in SimPlot recombination analysis for GX21122 and GX21123.

**Table 1 vetsci-10-00481-t001:** Primers used for CIAV DNA detection and whole genome amplification.

Primers	(Primer Sequence 5′ to 3′)	Length
CIAV ①-F	GACCGATCAACCCAAGCCTCC	1010 bp
CIAV ①-R	ATCTTCCCGGTCGCATAAGCA	
CIAV ②-F	CTTGCCGGTTCTTTAATCACCCT	751 bp
CIAV ②-R	CTCTTACCCAGCTGCCACACC	
CIAV ③-F	CTACATGGCAGCACCCGCATC	1055 bp
CIAV ③-R	TCCGGCACATTCTTA(G)AAACCAG	
CIAV ④-F	AATGAACGCTCTCCAAGAAG	540 bp
CIAV ④-R	AGCGGATAGTCATAGTAGAT	

**Table 2 vetsci-10-00481-t002:** Whole genome amplification reaction conditions.

Primers	Pre-Denatured	Denatured × 35	Annealing × 35	Extension × 35	Post Elongation
CIAV ①	95 °C, 5 min	94 °C, 30 s	61 °C, 30 s	72 °C, 70 s	72 °C, 7 min
CIAV ②	\	\	62 °C, 30 s	72 °C, 45 s	72 °C, 7 min
CIAV ③	\	\	58.7 °C, 30 s	72 °C, 70 s	72 °C, 7 min
CIAV ④	\	\	50.5 °C, 30 s	72 °C, 30 s	72 °C, 8 min

**Table 3 vetsci-10-00481-t003:** Recombination statistics of GX21122 and GX21123 using RDP 4.0.

Method	Recombination *p* Value
RDP	1.403 × 10^−2^
GENECONV	2.624 × 10^−2^
BootScan	3.598 × 10^−2^
MaxChi	4.106 × 10^−3^
Chimaera	1.071 × 10^−2^
SiScan	2.675 × 10^−4^
3Seq	6.864 × 10^−3^

Note: The recombination events showing *p* value < 0.05 were regarded as reliable.

## Data Availability

All data generated in the present study are available in the published manuscript.

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
