# Peer review of "Isolation, Identification, and Whole Genome Analysis of Chicken Infectious Anemia Virus in an Outbreak of Disease in Adult Layer Hens"

_vetsci, 2023, doi:10.3390/vetsci10070481_

Round 1

Reviewer 1 Report (Previous Reviewer 1)

The manuscript is well presented and the results are relevant, as there are descriptions in other parts of the world reporting the occurrence of virus genotypes with the same profile.

Author Response

Reviewer 2 Report (Previous Reviewer 3)

Dear authors,

following resubmission, the manuscript seems to have been improved enough to be considered for publication, at least in my opinion.

However, some issues still persist, including the following:

- There are some details provided in the Introduction that repeat or anticipate what is stated in the Materials and Methods regarding the clinical situation and diagnostic process. You should harmonize the  two sections to provide a coherent history of the outbreak and its investigation.

- At line 247, you stated that the culture experiment was repeated three times, but at line 399 it is stated that five attempts were made. Which is it?

- Line 268-9: Do you confirm that birds only died during the 7th and 14th, and not in other days of the first and second week?

- Line 435: I recommend replacing "genotype" with something like "genetic profile". With regards to CIAV, genotype means something very specific, which is not what is intended here.

- Appendix A. The column "Host" appears unnecessary (also "SPF" hosts, as stated for strain 20, are still chickens).

- Line 54: Replace "currently" with "to date".

- Line 55: "when infected with CIAV" is not needed.

- Line 66: "itself" is not needed.

- Line 141: Primers are plural.

- Line 253-256: These sentences need to be rephrased.

- Line 348: Replace "with" with "in".

- Line 365: the quotation marks are not needed.

- Line 54: Replace "currently" with "to date".

- Line 55: "when infected with CIAV" is not needed.

- Line 66: "itself" is not needed.

- Line 141: Primers are plural.

- Line 253-256: These sentences need to be rephrased.

- Line 348: Replace "with" with "in".

- Line 365: the quotation marks are not needed.

Author Response

Reviewer 3 Report (Previous Reviewer 2)

The authors have made a number of corrections and additions to the manuscript, which fully address the comments sent. I have only small comments with regard to the supplementary material, which is a duplication of Figure 2a-c, hence I consider it appropriate to remove it. In subsection 3.1, the first sentence appears to be incomplete and needs to be corrected. Also, there is an inconsistency in the manuscript with regard to the recombination site, i.e. in the results (line 349) the authors report the region 1679-2132 nt, while in the discussion (line 431) the position 1670-2130 is indicated. Please harmonise the demonstrated recombination region.

Author Response

Reviewer 4 Report (New Reviewer)

This manuscript, by Zeng et al, reported the isolation of three CIAV virus strains from a disease outbreak in a layer hen farm in China. Based on whole genome sequencing and phylogenetic analysis, all three virus isolates belong to genotype IIIIc. One of the three isolated virus strains, GX21121, exhibited high pathogenicity based on animal infection tests and histopathological examinations. In general, this report serves as an important reference for CIAV epidemiology studies in the future.  

The specific comments are:

1.       Line 21, “… subsequent animal infection experiments confirmed their high pathogenicity”: The pathogenicity was only tested for GX21121 strain. Please revise accordingly.

2.       Section 3.4, MDCC-MSB1 culture: It is great that negative results were shown in this report. I wonder if the cell adaption of the other two virus isolates has been tried or not? And are there any changes of the cell diameter and cell viability during the adaption for GX21121 strain at all?

3.       Line 257, “…the whole gene of the three CIAV strains was drawn (Fig.2d)”: I think Fig2d is only showing GX21121 strain. Please revise to ensure the consistency.

4.       Line 264: “(d)” is missing in the figure legend.

5.       Line 296: The figure legend for “s-u” is extremely long and some of the information can be put into the main text. Please revise accordingly.

6.       Line 456-457: Although the relationship between age of the chickens and the symptoms was discussed in the Discussion section, this study is not making a direct comparison of chicken of different ages, based on the data presented. I suggest deleting or rephrasing this sentence in the conclusion section.  

Author Response

This manuscript is a resubmission of an earlier submission. The following is a list of the peer review reports and author responses from that submission.

Round 1

Reviewer 1 Report

This manuscript is very interesting, because it is emerging some clinical cases involving CIAV in adult chickens, despite that this agent is commonly detected in young chicken. There are some points which the authors should add in the manuscript: please add the year of collection samples or emerging the outbreaks of CIAV (topic 2.1); topic 3.1 - please remove "clinical autopsy" and add "postmortem examination". Also, a macroscopy lesion should be presented in detail because it is very weak, as presented. The authors should add to the discussion some aspects of immunosuppression provoked by the agent, which could justify different clinical signs manifestation and lesions.

Reviewer 3 Report

In my opinion, the submitted manuscript cannot be taken into consideration for publication at the present state, primarily due to major formal flaws. Not only there are severe shortcomings in the quality of the English language hindering the comprehension of the conducted experiment, but several formal errors (missing/double spaces, captions cut in half, unnecessary headings, etc.) demand for a more thorough text revision prior to a possible resubmission.  

As for the scientific soundness, although the manuscript has its own merits (i.e. the study design appears quite exhaustive), there are some major flaws that need to be addressed and rectified. Some of them can be found in the list below:

- What is the reason of taking exactly six samples? Are they representing different houses/parts of a house/groups/etc.?

- More details could be provided regarding the outbreak, i.e. its course following the acute phase, the administered vaccination protocol, the results of other diagnostic investigations (if conducted) etc.

- The reason for trying to adapt the isolated virus to the MDCC-MSB1 appears unclear, considering that the failure to do so (which is not detrimental to the other results) is not mentioned in the discussion.

- In the case of GX21121, the phylogenetic analyses based on the whole genome and just the VP1 appear in conflict and do not support the conclusions reached by the authors. If the GX21121 evolved from the Del Ros vaccine solely through mutation (since no recombination was detected), this should be reflected even when considering just the VP1 (i.e. the two sequences should be somewhat related, even if dissimilar, whereas in the presented tree they are placed in two well-segregated branches. On the contrary, the overall topology of the two trees overlap quite nicely). My suggestion, even if it is difficult to say without knowing the position of the different primer pairs, would be to make sure that the obtained sequence was not obtained by mistakenly assembling genome portions of two coinfecting strains (a vaccine and a field one) together.    

- It is extremely hard to understand the soundness of the recombination analysis results. This is due to the fact that Figure 10 is not properly explained in the caption; in the first panel, there are many lines which are not explained in the respective legend (also, the image could be larger); in the second panel, I assume that the height is measured based on a reference which is not given; in the third panel, the y-axis should be limited to something like 0,9 to 1 to be readable.

- What is the reason behind choosing to isolate only one of the three strains? If the choice of considering GX21122 and GX21123 as having similar features could be justified by their high genetic homology, at least GX21121 should have been investigated separately.

Round 2

Reviewer 2 Report

No further comments

Author Response

Thanks for your advice and help.

Reviewer 3 Report

A significant improvement is evident in the quality of the English language, making the manuscript easier to read. Some inaccuracies are still present (missing letters in some of the squares of Figure 6; missing spaces; the unnecessary article often used before CIAV; etc.) but the most important points of concern are related to the robustness of the  phylogenetic analyses. I still believe that the discrepancy between the analyses of the full genome (closely resembling a vaccine) and of the sole VP1 (similar to other field strains belonging to a different genotype) of strain GX21121, which are possible albeit unlikely, were not discussed well enough. As for the recombination analyses, from what I can understand of the Figures (whose interpretation remains difficult), the evidence of recombination is quite unconvincing. In particular, strains GX21122 and GX21123 were proposed to have originated by separate recombination events. However, their VP1s (retrieved from GenBank) seem to be 99.78% identical with only one mismatch in the region in which recombination was proposed to have occured, making it unlikely that the two strains (found in the same farm) may have originated separately. In my opinion, the presented evidence is not enough to support the conclusions reached, so unfortunately I cannot endorse publication of the manuscript in its present form. 
